# Improving prediction of drug-target interactions based on fusing multiple features with data balancing and feature selection techniques

**Hakimeh Khojasteh**[1,2], **Jamshid Pirgazi**[2,3]*, **Ali Ghanbari Sorkhi**[3]

**1** Department of Computer Engineering, University of Zanjan, Zanjan, Iran, **2** School of Biological Sciences Institute for Research in Fundamental Sciences (IPM), Tehran, Iran, **3** Department of Computer Engineering, University of Science and Technology of Mazandaran, Behshahr, Iran

* j.pirgazi@mazust.ac.ir

**Data Availability Statement:** The source code of the method along with datasets is freely available at https://github.com/Khojasteh-hb/SRX-DTI.

## Abstract

Drug discovery relies on predicting drug-target interaction (DTI), which is an important challenging task. The purpose of DTI is to identify the interaction between drug chemical compounds and protein targets. Traditional wet lab experiments are time-consuming and expensive, that's why in recent years, the use of computational methods based on machine learning has attracted the attention of many researchers. Actually, a dry lab environment focusing more on computational methods of interaction prediction can be helpful in limiting search space for wet lab experiments. In this paper, a novel multi-stage approach for DTI is proposed that called SRX-DTI. In the first stage, combination of various descriptors from protein sequences, and a FP2 fingerprint that is encoded from drug are extracted as feature vectors. A major challenge in this application is the imbalanced data due to the lack of known interactions, in this regard, in the second stage, the One-SVM-US technique is proposed to deal with this problem. Next, the FFS-RF algorithm, a forward feature selection algorithm, coupled with a random forest (RF) classifier is developed to maximize the predictive performance. This feature selection algorithm removes irrelevant features to obtain optimal features. Finally, balanced dataset with optimal features is given to the XGBoost classifier to identify DTIs. The experimental results demonstrate that our proposed approach SRX-DTI achieves higher performance than other existing methods in predicting DTIs. The datasets and source code are available at: https://github.com/Khojasteh-hb/SRX-DTI.

## 1. Introduction

The main phase in the drug discovery process is to identify interactions between drugs and targets (or proteins), which can be performed by in vitro experiments. Identifying drug-target interaction plays a vital role in drug development that aims to identify new drug compounds for known targets and find new targets for current drugs [1,2]. The expansion of the human

**Funding:** The author(s) received no specific funding for this work.

**Competing interests:** The authors have declared that no competing interests exist.

genome project has provided a better diagnosis of disease, early detection of certain diseases, and identifying drug-target interactions (DTIs) [3]. Although significant efforts have been done in previous years, only a limited number of drug candidates have been permitted to reach the market by the Food and Drug Administration (FDA) whereas the maximum number of drug candidates have been rejected during clinical verifications, due to side effects or low efficacy [4]. Moreover, the cost of a new chemistry-based drug is often 2.6 billion dollars, and it takes typically 15 years to finish the drug development and approval procedure. This issue has been changing into a bottleneck to identifying the targets of any candidate drug molecules [2,5]. The experiment-based methods involve high cost, time-consuming, and small-scale limitations that motivate researchers to constantly develop computational methods for the exploitation of new drugs [2,6,7]. These computational methods offer a more efficient and cost-effective approach to drug discovery, allowing researchers to explore a larger range of potential drug candidates and predict their efficacy before investing significant resources into experimental testing. On the other side, the availability of online databases in this area, such as KEGG [8,9], DrugBank [10], PubChem [11], Davis [12], TTD [13,14], and STITCH [15] have been influencing Machine Learning (ML) researchers to develop high throughput computational methods.

Drug discovery involves identifying molecules that can effectively target and modulate the function of disease-related proteins. Besides developing computational methods for predicting drug-target interactions (DTIs), studying protein-protein interactions (PPIs) has also become a top priority for drug discovery, especially due to the SARS-CoV-2 pandemic [16–19]. Proteins are responsible for various essential processes in vivo via interactions with other molecules. Dysfunctional proteins are often responsible for diseases, making them crucial targets for the drug discovery process [20,21]. Abnormal PPIs can support the development of life-threatening diseases like cancer, further emphasizing the importance of identifying critical proteins and their interactions. Therefore, developing computational methods for identifying critical proteins in PPIs has become an important branch of drug discovery and treatment development [21,22]. In summary, understanding both DTIs and PPIs is critical for successful drug discovery. While this paper focuses on DTI prediction, it is important to consider PPI analysis as well in order to identify potential drug targets and improve the efficacy of drug development efforts.

The prior methods in DTI prediction can be mainly categorized into similarity-based methods and feature-based methods. In similarity-based methods, similar drugs or proteins are considered to find similar interaction patterns. These methods use many different similarity measures based on drug chemical similarity and target sequence similarity to identify drug-target interaction [23–25]. Feature-based methods consider drug–target interaction prediction as a binary classification problem and different classification algorithms such as Support Vector Machine (SVM) [26], random forest [27], rotation forest [28,29], XGBoost [30], and deep learning [31–35] have been employed to identify new interactions.

Various machine learning (ML) methods have been applied for drug-target prediction. Mousavian et al. utilized a support vector machine with features extracted from the Position Specific Scoring Matrix (PSSM) of proteins and molecular substructure fingerprint of drugs [26]. Shi et al. presented the LRF-DTIs method based on random forest, using pseudo-position specific scoring matrix (PsePSSM) and FP2 molecular fingerprint to extract features from proteins and drugs, and employing Lasso dimensionality reduction and Synthetic Minority Oversampling Technique (SMOTE) to handle unbalanced data [27]. Wang et al. proposed two methods based on Rotation Forest: RFDT, which used a PSSM descriptor and drug fingerprint as feature vectors [29], and RoFDT, which combined feature-weighted Rotation Forest (FwRF) with protein sequence encoded as PSSM, and drug structure fingerprints [28]. These

methods have shown promising results in predicting DTIs. Moreover, Mahmud et al. [30] proposed a computational model, called iDTi-CSsmoteB for the identification of DTIs. They utilized PSSM, amphiphilic pseudo amino acid composition (AM-PseAAC), and dipeptide PseAAC descriptors to present protein and molecular substructure fingerprint (MSF) to present drug molecule structure. Then, the oversampling SMOTE technique was applied to handle the imbalance of datasets, and the XGBoost algorithm as a classifier to predict DTIs.

The increase in the volume and diversity of data has led to the development of various deep learning platforms and libraries, such as DeepPurpose [32] and DeepDrug [35]. DeepPurpose [32] takes the SMILES format of the drug and amino acid sequence of the protein as input and transforms it into a specific format using a specific function. This format is then converted into a vector representation to be used in subsequent steps. This library provides eight encoders using different modalities of compounds, as well as utility functions to load pre-trained models and predict new drugs and targets. Yin et al. [35] proposed another deep learning framework called DeepDrug. Furthermore, variants of graph neural networks such as graph convolutional networks (GCNs) [35], graph attention networks (GATs) [36,37], and gated graph neural networks (GGNNs) [31,33,34] have been developed for DTI prediction.

We introduce SRX-DTI, a novel ML-based method for improving drug-target interaction prediction. First, we generate various descriptors for protein sequences, including Amino Acid Composition (AAC), Dipeptide Composition (DPC), Grouped Amino Acid Composition (GAAC), Dipeptide Deviation from Expected Mean (DDE), Pseudo Amino Acid Composition (PseAAC), Pseudo-Position-Specific Scoring Matrix (PsePSSM), Composition of K-spaced Amino Acid Group Pairs (CKSAAGP), Grouped Dipeptide Composition (GDPC), and Grouped Tripeptide Composition (GTPC). The drug is encoded as FP2 molecular fingerprint. Second, we use the technique namely Under Sampling by One-class Support Vector Machine (One-SVM-US) to balance the data, and the positive and negative samples are constructed using drug-target interaction information on the extracted features. Then, we perform the FFS-RF algorithm to select the optimal subset of features. Finally, after comparing various ML classifiers, we choose the XGBoost classifier to predict DTIs using 5-Fold cross-validation (CV). We evaluate the performance of our method using several metrics, including AUROC, AUPR, ACC, SEN, SPE, and F1-score. Our method achieves high AUROC values of 0.9920, 0.9880, 0.9788, and 0.9329 for EN, GPCR, IC, and NR, respectively. These results demonstrate that SRX-DTI outperforms existing methods for DTI prediction.

The rest of the paper is organized as follows: Materials and methods section describes the detail of the gold standard datasets, feature extraction, data balancing, and feature selection, we utilized in this paper. In the Results and discussion section, performance evaluation and experimental results are provided. Finally, the Conclusions section summarizes the conclusions.

## 2. Materials and methods

In this study, we propose a novel method of drug-target interaction prediction, which is called SRX-DTI. In the first step, drug chemical structures (SMILE format) and protein sequences (FASTA format) are collected from DrugBank and KEGG databases using their specific access IDs. In the next step, different feature extraction methods are applied to drug compounds and protein sequences to create a variety of features. Drug-target pair vectors are made based on known interactions and extracted features. Afterward, a balancing technique is utilized on DTI vectors to deal with imbalanced datasets, and drug–target features are selected through the FFS-RF to boost prediction performance. Finally, the XGBoost classifier is used on the balanced datasets with optimal features to predict DTIs. A schematic diagram of our proposed SRX-DTI model is shown in Fig 1.

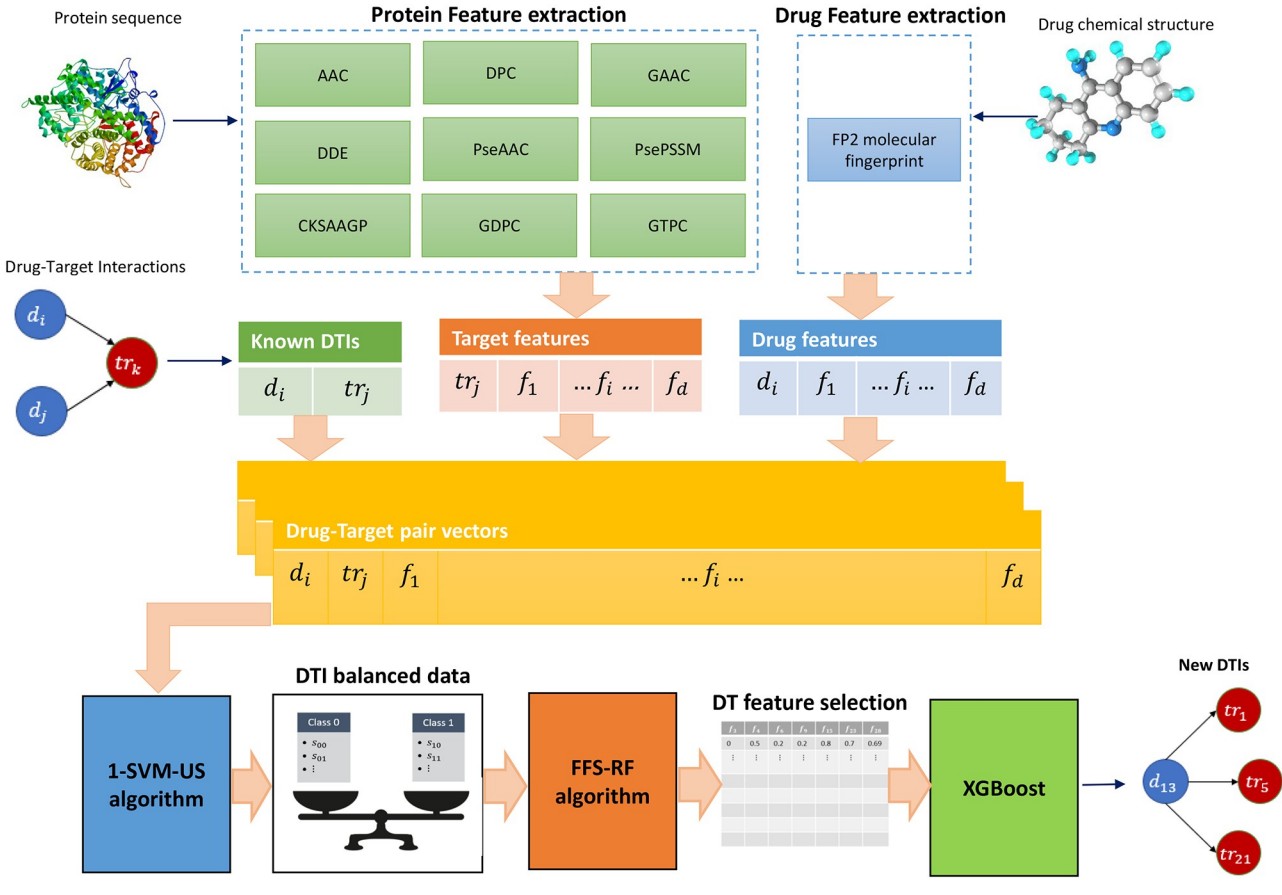

**Fig 1. The workflow of the proposed model to predict drug-target interactions.**

## 2.1 Drug–Target datasets

In this research, four golden standard datasets, including enzymes (EN), G-protein-coupled receptors (GPCR), ion channel (IC), and nuclear receptors (NR) released by Yamanishi et al. [38] are explored as benchmark datasets to evaluate the performance of the proposed SRX-DTI method in DTI prediction. All these datasets are freely available from http://web.kuicr.kyoto-u.ac.jp/supp/yoshi/drugtarget/. Yamanishi et al. [38] extracted information about drug-target interactions from DrugBank [39], KEGG [8,9], BRENDA [40], and SuperTarget [41]. The numbers of known interactions including enzymes, ion channels, GPCRs, and nuclear receptors are 2926, 1476, 635, and 90 respectively. The SRX-DTI model is also evaluated on the Davis Kinase binding affinity dataset [12]. The original Davis dataset represents 30,056 affinity bindings interactions between 442 proteins and 68 drug molecules. Here, we filter the dataset by removing all interactions with affinity $< 7$, resulting in the dataset used in this research. Finally, 2502 interactions are considered between proteins and drug molecules in the Davis dataset. A brief summary of these datasets is given in Table 1.

## 3. Feature extraction methods

In order to better identify drug-protein interactions, it seems advantageous to extract different features from drugs and targets. This allows us to have more complete information about the

**Table 1. Description of the gold standard datasets [12,38].**

| Datasets | Drugs | Targets | Interactions |
|---|---|---|---|
| EN | 445 | 664 | 2926 |
| GPCR | 223 | 95 | 635 |
| IC | 210 | 204 | 1476 |
| NR | 54 | 26 | 90 |
| Davis | 68 | 442 | 2502 (Affinity $\geq$ 7) |

known interactions and increase the detection rate. A brief summary of the ten groups of features is given in Table 2. Notice that there are two types of features. Drug related features and target related features in nine groups A, B, C, D, E, F, G, H, and I. In the following, these features are described, respectively. Whereas data diversity in the predictive models is very important, various subsets of these groups have been examined to select appropriate subsets. Based on drug and target descriptors, we constructed four subsets of features (AB, CD, EF, and GHI), which are given in Table 3. Also, notice that the drug features are coupled with singular target groups and these subsets. These four subsets have been selected to preserve certain properties of whole feature groups and at the same time, keep diversity in them.

## 3.1 Drug features

For drug compounds, different types of descriptors can be defined based on various types of drug properties such as FP2, FP3, FP4, and MACCS [42–44]. Some studies showed that these descriptors are molecular structure fingerprints that effectively represent the drug [27,45,46]. In this study, the FP2 format fingerprint is used to present drug compounds. This molecular fingerprint of the drug was extracted through these steps:

Step 1: For each drug, molecular structure as mol format is downloaded from the KEGG database (https://www.kegg.jp/kegg/drug/) by using its drug ID.

Step 2: The OpenBabel Software (available from http://openbabel.org/) is downloaded and installed.

Step 3: The drug molecules with mol file format are converted into the FP2 format molecular fingerprint using the OpenBabel software. The FP2 format molecular fingerprint is a

**Table 2. List of descriptors used in this study.**

| Descriptor | Number of Features | Feature Type | Feature Group |
|---|---|---|---|
| Molecular fingerprint | 256 | drug | |
| Amino acid composition (AAC) | 20 | target | A |
| Dipeptide composition (DPC) | 400 | target | B |
| Grouped amino acid composition (GAAC) | 5 | target | C |
| Dipeptide deviation from expected mean (DDE) | 400 | target | D |
| Pseudo amino acid composition (PseAAC) | 28 | target | E |
| Pseudo-position-specific scoring matrix (PsePSSM) | 220 | target | F |
| Composition of k-spaced amino acid group pairs (CKSAAGP) | 150 | target | G |
| Grouped dipeptide composition (GDPC) | 25 | target | H |
| Grouped tripeptide composition (GTPC) | 125 | target | I |

**Table 3. Four subsets of features based on drug and target descriptors.**

| Feature Combination | Number of Features |
|---|---|
| **AB** (with drug features) | 676 |
| **CD** (with drug features) | 661 |
| **EF** (with drug features) | 504 |
| **GHI** (with drug features) | 556 |

hexadecimal digit sequence of length 256 that is converted to a drug molecule 256-dimensional vector as a decimal digit sequence between 0 and 15.

## 3.2 Target features

**A) Amino acid composition (AAC):** The amino acid composition [47] is a vector of 20 dimensions, which calculates the frequencies of all 20 natural amino acids (i.e. "*ACDEFGHIKLMNPQRSTVWY*") as:

$$f_t = \frac{N(t)}{N}, \quad t \in \{A, C, D, \dots, Y\} \tag{1}$$

where $N(t)$ is the number of amino acid type $t$, while $N$ is the length of a protein sequence.

 **B) Dipeptide composition (DPC):** The Dipeptide Composition [48] gives 400 descriptors for protein sequence. It is calculated as:

$$D(r, s) = \frac{N_{rs}}{N - 1}, \quad t \in \{A, C, D, \dots, Y\} \tag{2}$$

where $N_{rs}$ is the number of dipeptides represented by amino acid types $r$ and $s$ and $N$ denotes the length of protein.

 **C) Grouped Amino Acid Composition (GAAC):** In the GAAC encoding [49], the 20 amino acid types are considered five classes according to their physicochemical properties. GAAC descriptor is the frequency of each amino acid group, which is calculated as:

$$f(g) = \frac{N(g)}{N}, \quad t \in \{g_1, g_2, g_3, g_4, g_5\} \tag{3}$$

$$N(g_t) = \sum N(t), \quad t \in g \tag{4}$$

where $N(g)$ is the number of amino acids in group $g$, $N(t)$ is the number of amino acid type $t$, and $N$ is the length of protein sequence.

 **D) Dipeptide Deviation from Expected mean (DDE):** The Dipeptide Deviation from Expected mean [48] is a feature vector, which is constructed by computing three parameters, i.e. dipeptide composition ($D_c$), theoretical mean ($T_m$), and theoretical variance ($T_v$). These three parameters and the DDE are defined as follows. $D_c(r, s)$, the dipeptide composition measure for the dipeptide '*rs*', is given as:

$$D_c(r, s) = \frac{N_{rs}}{N - 1}, \; r, s \in \{A, C, D, \dots, Y\} \tag{5}$$

where $N_{rs}$ is the number of dipeptides represented by amino acid types $r$ and $s$ and $N$ is the length of protein. $T_m(r, s)$, the theoretical mean, is given by:

$$T_m(r,s) = \frac{C_r}{C_N} \times \frac{C_s}{C_N} \qquad (6)$$

where $C_r$ is the number of codons, coding for the first amino acid, and $C_s$ is the number of codons, coding for the second amino acid in the given dipeptide 'rs' and $C_N$ is the total number of possible codons. $T_v(r, s)$, the theoretical variance of the dipeptide 'rs', is given by:

$$T_v(r,s) = \frac{T_m(r,s)(1 - T_m(r,s))}{N - 1} \qquad (7)$$

Finally, $DDE(r, s)$ is calculated as:

$$DDE(r,s) = \frac{D_c(r,s) - T_m(r,s)}{\sqrt{T_v(r,s)}} \qquad (8)$$

**E) Pseudo Amino Acid composition (PseAAC)**: To avoid completely losing the sequence-order information, the concept of PseAAC (pseudo amino acid composition) was proposed by Chou [50]; The idea of PseAAC has been widely used in bioinformatics including proteomics [51], system biology [52], such as predicting protein structural class [53], predicting protein subcellular localization [54], predicting DNA-binding proteins [55] and many other applications. In contrast with AAC which includes 20 components with each reflecting the occurrence frequency for One of the 20 native amino acids in a protein, the PseAAC contains a set of greater than 20 discrete factors, where the first 20 represent the components of its conventional amino acid composition while the additional factors are a series of rank-different correlation factors along a protein chain. According to the concept of PseAAC [50], any protein sequence formulated as a PseAAC vector given by:

$$x = \left[x_1, \ x_2, \ \ldots, \ x_{19}, \ x_{20}, \ x_{20+1}, \ \ldots, \ x_{20+\lambda}\right]^T, (\lambda < L) \qquad (9)$$

where $L$ is the length of protein sequence, and $\lambda$ is the sequence-related factor that choosing a different integer for, will lead to a dimension-different PseAAC. Each of the components can be defined as follows:

$$x_u = \begin{cases} \dfrac{f_i}{\sum_{i=1}^{20} f_i + w \sum_{k=1}^{\lambda} \tau_k}, & 1 \le u \le 20 \\[3ex] \dfrac{w\tau_{u-20}}{\sum_{i=1}^{20} f_i + w \sum_{k=1}^{\lambda} \tau_k}, & 20 + 1 \le u \le 20 + \lambda \end{cases} \qquad (10)$$

where $w$ is the weight factor, and $f_i$ indicates the frequency at $i - th$ AA in protein sequence. The $\tau_k$, the $k$-th tier correlation factor reflects the sequence order correlation between all the $k$-th most contiguous residues as formulated by:

$$\tau_k = \frac{1}{L - K} \sum_{i=1}^{L-K} J_{i,i+k}, K < L \qquad (11)$$

with

$$J_{i,i+k} = \frac{1}{\Gamma} \sum_{q-1}^{\Gamma} \left[\Phi_q\left(R_{i+k}\right) - \Phi_q(R_i)\right]^2 \qquad (12)$$

where $\Phi_q(R_i)$ is the $q$-th function of amino acid $R_i$, and $\Gamma$ is the total number of the functions considered. In this research, the protein functions which are considered, includes hydrophobicity value, hydrophilicity value, and side chain mass of amino acid. Therefore, the total number of functions $\Gamma$ is 3.

In this study, $\lambda$ is set to 1 and $W$ is set to 0.05. The output characteristic dimensions of each target protein are 28 for the PseAAC descriptor.

**F) Pseudo position specific scoring matrix (PsePSSM)**: To represent characteristics of the amino acid (AA) sequence for protein sequences, the pseudo-position specific scoring matrix (PsePSSM) features introduced by Shen et al. [56] are used. The pseudo-position specific scoring matrix (PsePSSM) features encode the protein sequence's evolution and information which have been broadly used in bioinformatics research [16,56,57].

For each target sequence P with L amino acid residues, PSSM is used as its descriptor proposed by Jones et al. [58]. The position-specific scoring matrix (PSSM) with a dimension of L×20 can be defined as:

$$P_{PSSM} = \begin{bmatrix} M_{1\to1} & M_{1\to2} & \cdots & M_{1\to20} \\ M_{2\to1} & M_{2\to2} & \cdots & M_{2\to20} \\ \vdots & \vdots & \vdots & \vdots \\ M_{i\to1} & M_{i\to2} & \cdots & M_{i\to20} \\ \vdots & \vdots & \vdots & \vdots \\ M_{L\to1} & M_{L\to2} & \cdots & M_{L\to20} \end{bmatrix} \tag{13}$$

where $M_{i,j}$ indicates the score of the amino acid residue in the $i$th position of the protein sequence being mutated to amino acid type $j$ during the evolution process. Here, for simplifying the formulation, it is used the numerical codes 1, 2,..., 20 to represent the 20 native amino acid types according to the alphabetical order of their single character codes. It can be searched using the PSI-BLAST [59] in the Swiss-Prot database. A positive score shows that the corresponding residue is mutated more frequently than expected, and a negative score is just the contrary.

In this work, the parameters of PSI-BLAST are set as the threshold of E-value equals 0.001, the maximum number of iterations for multiple searches equals 3, and the rest of the parameters by default. Each element in the original PSSM matrix was normalized to the interval (0, 1) using Eq (14):

$$\overline{M}_{i\to j} = \frac{1}{1 + \exp\left(-M_{i\to j}\right)} \tag{14}$$

However, due to different lengths in target sequences, making the PSSM descriptor as a uniform representation can be helpful, one possible representation of the protein sample P is:

$$\overline{P}_{PSSM} = \left[\overline{M}_1, \overline{M}_2, \ldots, \overline{M}_{20}\right] \tag{15}$$

where T is the transpose operator, and

$$\overline{M}_j = \frac{1}{L}\sum_{i=1}^{L} M_{i\to j}(j = 1, 2, \ldots, 20) \tag{16}$$

where $M_{i\to j}$ is the average score of the amino acid residues in the protein P changed to the $j$th amino acid residue after normalization, $M_j$ represents the average score of the amino acid residue in protein P being mutated amino acid type j during the process of evolution. However, if

$P_{PSSM}$ of Eq (13) represents the protein P, all the sequence-order information would be lost. To avoid complete loss of the sequence-order information, the concept of the pseudo amino acid composition introduced by Chou [60], i.e. instead of Eq (11), we use position-specific scoring matrix (PsePSSM) to represent the protein P:

$$P_{psePSSM}^{\lambda} = \left[ \overline{M}_1, \overline{M}_2, \ldots, \overline{M}_{20}, G_1^1, G_2^1, \ldots, G_{20}^1, G_1^{\lambda}, G_2^{\lambda}, \ldots, G_{20}^{\lambda} \right]^T \tag{17}$$

where

$$G_j^{\lambda} = \frac{1}{L - \lambda} \sum_{i=1}^{L-\lambda} \frac{\left[ \overline{M}_{i \to j} - \overline{M}_{(i+\lambda) \to j} \right]^2}{(j = 1, 2, \ldots, 20; 0 \le \lambda \le L)} \tag{18}$$

where $G_j^{\lambda}$ represents the correlation factor of the j - th amino acid and $\lambda$ is the continuous distance along the protein sequence. This means that $G_j^1$ is the relevant factor coupled along the most continuous PSSM score on the protein chain of amino acid type $j$, $G_j^2$ is the second closest PSSM score by coupling, and so on. Therefore, a protein sequence can be defined as Eq (15) using PsePSSM and produces a $20 + 20 \times \lambda\text{-}dimensional$ feature vector. In this study, $\lambda$ is set to 10. The output characteristic dimension of each target protein is 220 for the PsePSSM descriptor.

**G) Composition of k-spaced amino acid group pairs (CKSAAGP)**: The Composition of k-Spaced Amino Acid Group Pairs (CKSAAGP) [61] defines the frequency of amino acid group pairs separated by any k residues (the default maximum value of k is set as 5). If $k = 0$, the 0-spaced group pairs are represented as:

$$\left( \frac{N_{g1g1}}{N_{total}}, \frac{N_{g1g2}}{N_{total}}, \frac{N_{g1g3}}{N_{total}}, \ldots, \frac{N_{g5g5}}{N_{total}} \right)_{25} \tag{19}$$

where the value of each descriptor indicates the composition of the corresponding residue group pair in a protein sequence. For a protein of length $P$ and $k = 0, 1, 2, 3, 4$ and $5$, the values of $N_{total}$ are $P—1, P—2, P—3, P—4, P—5$ and $P—6$ respectively.

**H) Grouped dipeptide composition (GDPC)**: The Grouped Di-Peptide Composition encoding [61] is a vector of 25 dimensions, which is another variation of the DPC descriptor. It is defined as:

$$f(r, s) = \frac{N_{rs}}{N - 1}, r, s \in \{g_1, g_2, g_3, g_4, g_5\} \tag{20}$$

where $N_{rs}$ is the number of dipeptides represented by amino acid types $r$ and $s$ and $N$ denotes the length of a protein.

**I) Grouped tripeptide composition (GTPC)**: The Grouped Tri-Peptide Composition encoding [61] is also a variation of the TPC descriptor, which generates a vector of 125 dimensions, defined as:

$$f(r, s) = \frac{N_{rst}}{N - 2}, r, s, t \in \{g_1, g_2, g_3, g_4, g_5\} \tag{21}$$

where $N_{rst}$ is the number of tripeptides represented by amino acid types $r$, $s$ and $t$. $N$ denotes the length of a protein.

```
Algorithm 1. UnderSampling by One-class SVM (One-SVM-US).
1: n_minority ← number of minority class samples
2: n_majority ← number of majority class samples
3: df [1.... n_majority] ← Majority class Samples
4: df [1.... n_minority] ← Minority class Samples
```

```
5: Model ← OneClassSVM(df [1... .n_majority]) // One-class SVM with RBF
kernel and
6:                                          // γ = 1/n_majority
7: scores ←— Model.makeDecision(df [1... .n_majority])
8: Q ←— max_scores decisionFunction(scores)
9: outlierScores ←— Q-scores
10: sortedScores ←— sort(outlierScores)
11: SelectedIndices = sortedScores [1... .n_minority]
12: X1 = {}
13: for each index ∈ selectedIndices do
14: X1 = X1 ∪ df [index]
15: endfor
16: FinalData = X1 ∪ df [1... .n_minority]
```

## 4. Data balancing technique

The experiment datasets that we used in this study were highly imbalanced. Imbalanced datasets can present a challenge for many machine learning algorithms, as they may prioritize the majority class and ignore the minority class, leading to poor performance on the minority class. Different techniques have been utilized to balance the imbalanced dataset, such as random undersampling [26,62,63], cluster undersampling [64,65], and SMOTE technique [27,30]. To address the issue of imbalanced data in our study, we developed a new undersampling algorithm called One-SVM-US, which uses One-class Support Vector Machine (SVM) to deal with imbalanced data. The steps of the One-SVM-US algorithm were implemented as Algorithm 1. In the first step, the known DTIs are considered positive samples. For enzymes, ion channels, GPCRs, nuclear receptors, and the Davis dataset, the number of positives is 2926, 1476, 635, 90, and 2502, respectively. In the next step, the algorithm considers all of the possible interactions in five datasets as negative samples except the ones that have been known as positive. By performing the One-SVM-US algorithm, it would result in a balanced dataset with equal numbers of positive and negative samples.

A One-Class Support Vector Machine (One-class SVM) [66], is a semi-supervised global anomaly detector. This algorithm needs a training set that contains only one class. The One-SVM-US technique based on One-class SVM considers all possible combinations of drug and target by discarding those that are positive samples. This algorithm uses a hypersphere to encompass all of the instances instead of using a hyperplane to separate two classes of samples. We apply the RBF kernel for SVM. The setting for the parameter γ was investigated, which was the simple heuristic γ = 1/no. of data points. To compute the outlier score, first, the maximum value of the decision function is obtained by:

$$Q = \max_{x} \text{decision\_function}(x')  \qquad (22)$$

where $x$ refers to the vector of scores. Then, we obtained the outlier score as follows:

$$\text{outlier\_scores} = Q - \text{decision\_function}(x)  \qquad (23)$$

Then, the outlier scores are sorted in ascending and the $n_{minority}$ samples are selected from the sorted list. The final data is constructed from the combination of the minority class from the original experimental dataset and the majority class chosen by the proposed method. Even though, we would like to mention that Algorithm 1 performs effectively to make balanced datasets.

## 5. Feature selection technique

Considering that reducing the number of input features can lead to both reducing the computational cost of modeling and, in some cases, improving the performance of the model. We develop a feature selection algorithm with RF, called FFS-RF. This algorithm was developed and implemented based on the forward feature selection (FFS) technique [67] that coupled with RF to obtain optimal features in DTI. The RF approach [68] is an ensemble method that combines a large number of individual binary decision trees. The performance of the RF model in feature selection was evaluated by a 5-fold CV to construct an effective prediction framework. Forward feature selection is an iterative process, which begins with an empty set of features. After each iteration, it keeps adding on a feature and evaluates the performance to check whether it is improving the performance or not. The FFS-RF technique continues until the addition of a new feature does not improve the performance of the model, as outlined in Algorithm 2 step by step.

```
Algorithm 2. Forward Feature Selection algorithm with RF (FFS-RF).
 1: FS⁰ = ∅
 2: F⁰ = {f₁, f₂, ..., fₙ}
 3: i = 0
 4: opt = 0
 5: iter = 0
 6: while (i < n)
 7:   k = size (F⁽ⁱ⁾)
 8:   max = 0
 9:   feature = 0
10:   for j from 1 to k
11:     score = eval (F⁽ⁱ⁾ⱼ)
12:     if (score > max)
13:       max = score
14:       feature = F⁽ⁱ⁾ⱼ
15:     endif
16:   endfor
17:   if (max > opt)
18:       opt = max
19:       iter = i
20:   endif
21:     FS⁽ⁱ⁺¹⁾ = F⁽ⁱ⁾ + feature
22:     F⁽ⁱ⁺¹⁾ = F⁽ⁱ⁾ − feature
23:     i++
24: endwhile
```

## 6. Results and discussion

In this section, we explain the experimental results of our proposed method in DTI prediction. We implemented all the phases, i.e., features extraction, data balancing, and classifiers of the proposed model in Python language (Python 3.10 version) using the Scikit-learn library. Some of the target descriptors were calculated by the iFeature package [61] and the rest of them were implemented in Python language. OpenBabel Software was used to extract fingerprint descriptors from drugs. All of the implantations were performed on a computer with a processor 2.50 GHz Intel Xeon Gold 5–2670 CPU and 64 GB RAM.

### 6.1 Performance evaluation

Most of the methods in DTI prediction [5,6,26,30] have utilized 5-fold cross validation (CV) to assess the power of the model to generalize. We also use the 5-fold CV to estimate the skill of

the SRX-DTI model on new data and make a fair comparison with the other state-of-the-art methods. The drug–target datasets were split into 5 subsets where each subset was used as a testing set. In the first iteration, the first subset is used to test the model and the rest are used to train the model. In the second iteration, 2nd subset is used as the testing set while the rest serves as the training set. This process is repeated until each fold of the 5 folds is used as the testing set. Then, the performance is reported as the average of the five validation results for drug-target datasets.

In this study, we perform three types of analyses. First, the importance of feature extraction is discussed. Secondly, we investigate the impact of our balancing technique (One-SVM-US) versus the random undersampling technique on CV results. Finally, the effectiveness of the feature selection method is analyzed.

We used the following evaluation metrics to assess the performance of the proposed model: accuracy (ACC), sensitivity (SEN), specificity (SPE), and F1 Score.

$$ACC = \frac{TP + TN}{TP + FP + TN + FN} \tag{24}$$

$$SEN = \frac{TP}{TP + FN} \tag{25}$$

$$SPE = \frac{TN}{TN + FP} \tag{26}$$

$$F1 = \frac{2TP}{2TP + FP + FN} \tag{27}$$

where based on four metrics, namely true positives (TP), false positives (FP), true negatives (TN), and false negatives (FN) are to present an overview of performance. Moreover, we used AUROC (Area Under Receiver Operating Characteristic curve) to show the power of discrimination of the model between the positive class and the negative class. The AUPR (Area Under Precision Recall curve) was also used which would be more informative when there is a high imbalance in the data [69].

## 6.2 The effectiveness of feature groups

We constructed nine different feature groups namely A, B, C, D, E, F, G, H, and I, which all were coupled with drug features to assess the effects of the different sets of features on the performance of the different classifiers including SVM, RF, MLP, and XGBoost. The feature groups have already been reported in Table 2. We also created some subsets from the groups (AB, CD, EF, and GHI), which are given in Table 3. The selection of the best combination can be considered an optimization problem. Here, we combine feature descriptors based on non-monotonic information and the performance results we get for different classifiers in single feature groups.

We performed experiments to test the effectiveness of the feature groups. In the experiments, we changed the feature groups and applied the random undersampling technique to balance datasets. Statistics of the prediction performance for different classifier models are given in Tables 4 and 5.

Focus on the EN dataset, we compared the DTI prediction performance of four different classifiers on nine feature groups and four subsets of them. We also highlighted several possible characteristics that could be considered to select the best classifier in DTI prediction. The

**Table 4. Performance of Support Vector Machine, Random Forest, Multilayer perception, and XGBoost classifiers on the gold standard datasets using different feature group combinations and random undersampling technique.**

| Dataset | Feature Combination | Classifier | AUROC | AUPR | ACC | SEN | SPE | F1 |
|---|---|---|---|---|---|---|---|---|
| EN | A | SVM | 0.8687 | 0.8642 | 0.7771 | 0.7457 | 0.8085 | 0.7700 |
| | | RF | 0.8050 | 0.8042 | 0.7242 | 0.6382 | 0.8103 | 0.6984 |
| | | MLP | 0.8939 | 0.8933 | 0.8335 | 0.8635 | 0.8034 | 0.8384 |
| | | XGBoost | **0.9253** | **0.9265** | **0.8565** | **0.8447** | **0.8684** | **0.8549** |
| | B | SVM | 0.9135 | 0.9019 | 0.8429 | 0.8396 | 0.8462 | 0.8425 |
| | | RF | 0.8109 | 0.8277 | 0.7378 | 0.6365 | 0.8393 | 0.7085 |
| | | MLP | 0.9391 | 0.9378 | 0.8702 | 0.8976 | 0.8427 | 0.8738 |
| | | XGBoost | **0.9271** | **0.9254** | **0.8531** | **0.8601** | **0.8462** | **0.8542** |
| | C | SVM | 0.8457 | 0.8521 | 0.7763 | 0.7457 | 0.8068 | 0.7694 |
| | | RF | 0.7832 | 0.7840 | 0.7037 | 0.5939 | 0.8137 | 0.6673 |
| | | MLP | 0.8573 | 0.8496 | 0.7925 | 0.8242 | 0.7607 | 0.7990 |
| | | XGBoost | **0.9071** | **0.9064** | **0.8386** | **0.8362** | **0.8410** | **0.8383** |
| | D | SVM | 0.9164 | 0.9042 | 0.8454 | 0.8447 | 0.8462 | 0.8454 |
| | | RF | 0.7761 | 0.7949 | 0.7054 | 0.6109 | 0.8000 | 0.6748 |
| | | MLP | 0.9385 | 0.9331 | 0.8702 | 0.9027 | 0.8376 | 0.8744 |
| | | XGBoost | **0.9307** | **0.9326** | **0.8642** | **0.8584** | **0.8701** | **0.8635** |
| | E | SVM | 0.8588 | 0.8621 | 0.7788 | 0.7474 | 0.8103 | 0.7718 |
| | | RF | 0.8004 | 0.8070 | 0.7208 | 0.6416 | 0.8000 | 0.6969 |
| | | MLP | 0.8970 | 0.8954 | 0.8412 | 0.8567 | 0.8256 | 0.8437 |
| | | XGBoost | **0.9327** | **0.9348** | **0.8634** | **0.8652** | **0.8615** | **0.8637** |
| | F | SVM | 0.8777 | 0.8708 | 0.8155 | 0.7628 | 0.8684 | 0.8054 |
| | | RF | 0.7972 | 0.8052 | 0.7233 | 0.5751 | 0.8718 | 0.6754 |
| | | MLP | 0.9156 | 0.9083 | 0.8352 | 0.9044 | 0.7658 | 0.8460 |
| | | XGBoost | **0.9368** | **0.9356** | **0.8745** | **0.8737** | **0.8752** | **0.8745** |
| | G | SVM | 0.8946 | 0.8972 | 0.8155 | 0.7986 | 0.8325 | 0.8125 |
| | | RF | 0.8096 | 0.8176 | 0.7319 | 0.6672 | 0.7966 | 0.7135 |
| | | MLP | 0.9254 | 0.9202 | 0.8736 | 0.8754 | 0.8718 | 0.8739 |
| | | XGBoost | **0.9317** | **0.9322** | **0.8599** | **0.8652** | **0.8547** | **0.8608** |
| | H | SVM | 0.8645 | 0.8741 | 0.7797 | 0.7457 | 0.8137 | 0.7721 |
| | | RF | 0.8020 | 0.7990 | 0.7293 | 0.6126 | 0.8462 | 0.6937 |
| | | MLP | 0.9007 | 0.8999 | 0.8215 | 0.8379 | 0.8051 | 0.8245 |
| | | XGBoost | **0.9233** | **0.9216** | **0.8488** | **0.8447** | **0.8530** | **0.8483** |
| | I | SVM | 0.8931 | 0.8953 | 0.8079 | 0.7833 | 0.8325 | 0.8031 |
| | | RF | 0.8132 | 0.8112 | 0.7455 | 0.6485 | 0.8427 | 0.7183 |
| | | MLP | 0.9203 | 0.9092 | 0.8676 | 0.8805 | 0.8547 | 0.8694 |
| | | XGBoost | **0.9235** | **0.9234** | **0.8497** | **0.8464** | **0.8530** | **0.8493** |

results indicated that XGBoost is competitive in predicting interactions. We also made some subsets from single groups namely: AB, CD, EF, and GHI. Two classifiers include MLP and XGBoost had close performance and outperforms other ML methods to predict DTIs.

## 6.3 The influence of the data balancing techniques

Imbalanced data classification is a significant challenge for predictive modeling. Most of the machine learning algorithms used for classification were designed around the assumption of an equal number of samples for each class. Imbalanced data lead to biased prediction results in

**Table 5. Performance of Support Vector Machine, Random Forest, Multilayer perception, and XGBoost classifiers on the gold standard datasets using different subsets of feature groups combinations and random undersampling technique.**

| Dataset | Feature Combination | Classifier | AUROC | AUPR | ACC | SEN | SPE | F1 |
|---------|---------------------|-----------|-------|------|-----|-----|-----|-----|
| **EN** | AB | SVM | 0.9133 | 0.9023 | 0.8429 | 0.8396 | 0.8462 | 0.8425 |
| | | RF | 0.8092 | 0.8145 | 0.7404 | 0.6280 | 0.8530 | 0.7077 |
| | | MLP | **0.9395** | **0.9373** | **0.8779** | **0.8857** | **0.8701** | **0.8789** |
| | | XGBoost | **0.9277** | **0.9290** | **0.8599** | **0.8618** | **0.8581** | **0.8603** |
| | CD | SVM | 0.9162 | 0.9039 | 0.8454 | 0.8447 | 0.8462 | 0.8454 |
| | | RF | 0.7746 | 0.7881 | 0.7096 | 0.6297 | 0.7897 | 0.6846 |
| | | MLP | **0.9406** | **0.9387** | **0.8719** | **0.8908** | **0.8530** | **0.8744** |
| | | XGBoost | **0.9328** | **0.9379** | **0.8659** | **0.8652** | **0.8667** | **0.8659** |
| | EF | SVM | 0.8855 | 0.8798 | 0.8155 | 0.7628 | 0.8684 | 0.8054 |
| | | RF | 0.7997 | 0.8115 | 0.7190 | 0.5597 | 0.8786 | 0.6660 |
| | | MLP | **0.9308** | **0.9315** | **0.8591** | **0.8925** | **0.8256** | **0.8637** |
| | | XGBoost | **0.9397** | **0.9411** | **0.8779** | **0.8857** | **0.8701** | **0.8789** |
| | GHI | SVM | 0.9051 | 0.9020 | 0.8318 | 0.8157 | 0.8479 | 0.8291 |
| | | RF | 0.8201 | 0.8351 | 0.7592 | 0.7321 | 0.7863 | 0.7526 |
| | | MLP | **0.9278** | **0.9189** | **0.8668** | **0.8805** | **0.8530** | **0.8687** |
| | | XGBoost | **0.9288** | **0.9316** | **0.8531** | **0.8618** | **0.8444** | **0.8545** |

ML problems. The drug–target datasets are highly imbalanced. The number of known DTI (positive samples) is significantly smaller than that of unknown DTI (negative samples), which causes to achieve poor performance results of the prediction model. To make balancing in datasets, we used the One-SVM-US technique to build a powerful model. Here, we make experiments to compare the One-SVM-US technique and random undersampling technique to balance datasets in the model. The experimental results are shown in Tables 6–8, which reveal the efficiency of the One-SVM-US algorithm.

We observe from Table 6 that the model performance on balanced with Random undersampling and balanced with One-SVM-US in group AB. The results show a significant preference for the AUROC, AUPR, ACC, SEN, SPE, and F1 evaluation metrics by applying One-SVM-US. For the EN dataset, the model achieved AUROC values of 0.9920 in One-SVM-US, and 0.8753 in Random undersampling. In the case of the GPCR dataset, the model obtained AUROC values of 0.9880 and 0.7866, in One-SVM-US and Random undersampling, respectively. For the IC dataset, the model yielded an AUROC of 0.9788 in One-SVM-US and 0.8513 in Random undersampling. Similarly, AUROC values of the model using NR data are 0.9329 in One-SVM-US and 0.6496 in Random undersampling.

**Table 6. Comparison of prediction results on balanced with Random undersampling and balanced with One-SVM-US in group AB.**

| Dataset | Sampling method | AUROC | AUPR | ACC | SEN | SPE | F1 |
|---------|-----------------|-------|------|-----|-----|-----|-----|
| **EN** | **Random undersampling** | 0.8753 | 0.8625 | 0.8006 | 0.8491 | 0.7887 | 0.8196 |
| | **One-SVM-US** | 0.9920 | 0.9975 | 0.9901 | 0.9947 | 0.9967 | 0.9956 |
| **GPCR** | **Random undersampling** | 0.7866 | 0.7728 | 0.7354 | 0.7907 | 0.7360 | 0.7727 |
| | **One-SVM-US** | 0.9880 | 0.9940 | 0.9732 | 0.9658 | 1.0000 | 0.9826 |
| **IC** | **Random undersampling** | 0.8513 | 0.8289 | 0.7873 | 0.8447 | 0.7730 | 0.8233 |
| | **One-SVM-US** | 0.9788 | 0.9863 | 0.9543 | 0.9429 | 0.9743 | 0.9565 |
| **NR** | **Random undersampling** | 0.6496 | 0.6115 | 0.6556 | 0.9231 | 0.7826 | 0.8000 |
| | **One-SVM-US** | 0.9329 | 0.9407 | 0.8611 | 1.0000 | 0.8889 | 0.9474 |

**Table 7. Comparison of prediction results on balanced with Random undersampling and balanced with One-SVM-US in group EF.**

| Dataset | Sampling method | AUROC | AUPR | ACC | SEN | SPE | F1 |
|---------|-----------------|-------|------|-----|-----|-----|----|
| EN | Random undersampling | 0.9024 | 0.9008 | 0.8271 | 0.8737 | 0.8286 | 0.8506 |
|    | One-SVM-US | 0.9910 | 0.9964 | 0.9766 | 0.9647 | 0.9934 | 0.9785 |
| GPCR | Random undersampling | 0.8236 | 0.7762 | 0.7740 | 0.8527 | 0.6800 | 0.7885 |
|      | One-SVM-US | 0.9881 | 0.9928 | 0.9638 | 0.9487 | 0.9562 | 0.9487 |
| IC | Random undersampling | 0.8765 | 0.8527 | 0.8150 | 0.8479 | 0.8191 | 0.8424 |
|    | One-SVM-US | 0.9796 | 0.9880 | 0.9580 | 0.9643 | 0.9807 | 0.9712 |
| NR | Random undersampling | 0.7781 | 0.7029 | 0.7278 | 0.9231 | 0.7391 | 0.7742 |
|    | One-SVM-US | 0.8837 | 0.9033 | 0.8000 | 1.0000 | 0.7778 | 0.9000 |

There is a similar pattern in group EF, which is shown in Table 7. In the case of EN, the prediction results of ACC, SEN, SPE, and F1 on balanced data with One-SVM-US are 0.9901, 0.9947, 0.9967, and 0.9956, which are 0.1895, 0.1456, 0.208, and 0.176 higher than those balanced with Random undersampling, respectively. These prediction results show that the One-SVM-US technique obtains a comparatively advantageous performance. In the case of GPCR, IC, and NR datasets, the ACC, SEN, SPE, and F1 results for balanced data with One-SVM-US and balanced with Random undersampling are in Table 6. The values of these metrics are also shown in Table 7 for group EF. To better analyze the proposed methods, the ROC curves of two data balancing techniques are shown in Fig 2a–2d. These curves demonstrate discriminative ability in group AB, the ROC curve using the One-SVM-US covers the largest area, which is higher than the Random undersampling. The ROC curves of group EF are also shown in Fig 3a–3d, which also cover the larger area in the One-SVM-US technique in comparison with the Random undersampling technique.

We can see from Table 8 that the model performance on balanced with Random undersampling and balanced with One-SVM-US on the Davis dataset. It can be observed that the proposed One-SVM-US exhibits a similar performance in all datasets. For the Davis dataset, the model AUROC values are 0.9786, 0.9839, 0.9756, and 0.9696 in groups AB, CD, EF, and GHI, respectively. For each feature group, the One-SVM-US technique performs better in terms of AUPR 0.9848, 0.9896, 0.9835, and 0.9781 for groups AB, CD, EF, and GHI, respectively. These results demonstrate that the balanced dataset using One-SVM-US significantly outperforms the balanced dataset using Random undersampling in the case of ROC curves. The accuracy of the XGBoost classifier has been improved after utilizing the One-SVM-US. For all five datasets on the SEN, SPE, and F1 metrics, the results are significantly better in

**Table 8. Comparison of prediction results on balanced with Random undersampling and balanced with One-SVM-US on Davis dataset.**

| Groups | Sampling method | AUROC | AUPR | ACC | SEN | SPE | F1 |
|--------|-----------------|-------|------|-----|-----|-----|----|
| AB | Random undersampling | 0.8566 | 0.8287 | 0.7932 | 0.8182 | 0.8058 | 0.8182 |
|    | One-SVM-US | 0.9786 | 0.9848 | 0.9384 | 0.9256 | 0.9555 | 0.9382 |
| CD | Random undersampling | 0.8812 | 0.8800 | 0.8054 | 0.8337 | 0.8161 | 0.8312 |
|    | One-SVM-US | 0.9839 | 0.9896 | 0.9474 | 0.9483 | 0.9768 | 0.9613 |
| EF | Random undersampling | 0.8898 | 0.8847 | 0.8098 | 0.8085 | 0.8079 | 0.8132 |
|    | One-SVM-US | 0.9756 | 0.9835 | 0.9329 | 0.9153 | 0.9478 | 0.9287 |
| GHI | Random undersampling | 0.8703 | 0.8543 | 0.8000 | 0.8298 | 0.8058 | 0.8250 |
|     | One-SVM-US | 0.9696 | 0.9781 | 0.9207 | 0.8967 | 0.9807 | 0.9353 |

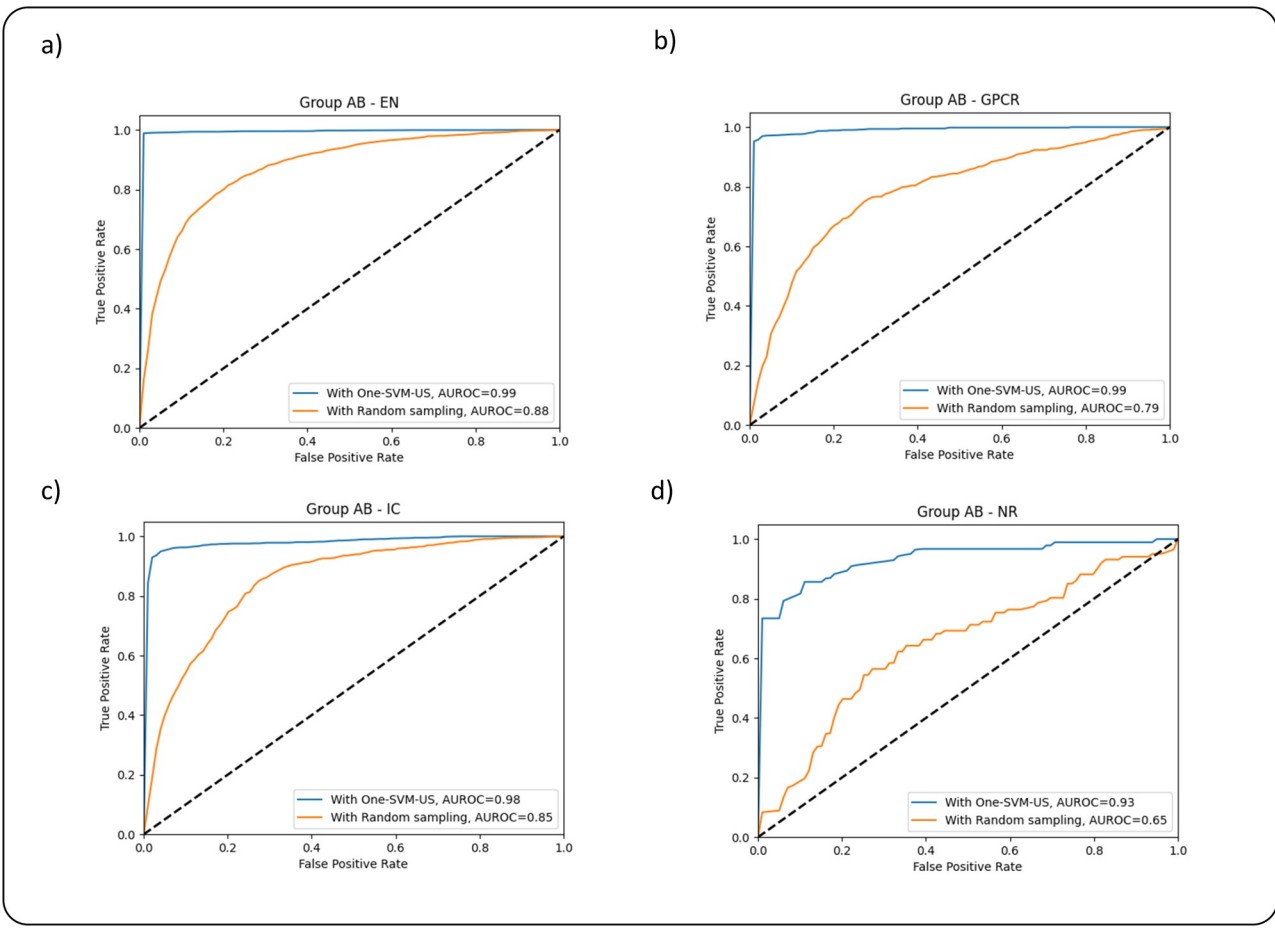

**Fig 2. ROC curves of the feature group AB using Random undersampling and One-SVM-US techniques on the datasets: (a) EN, (b) GPCR, (c) IC, and (d) NR.**

One-SVM-US. Ultimately, One-SVM-US is the efficient method to make balanced datasets to reduce bias and boost the model's performance.

## 6.4 The effectiveness of feature selection technique

Feature selection is extremely important in ML because it primarily serves as a fundamental technique to direct the use of informative features for a given ML algorithm. Feature selection techniques are especially indispensable in scenarios with many features, which is known as the curse of dimensionality. The solution is to decrease the dimensionality of the feature space via a feature selection method.

A feature selection technique by selecting an optimal subset of features reduces the computational cost. Various feature selection techniques have been utilized in DTI prediction [1,6,64]. The wrapper-based methods refer to a category of supervised feature selection methods that uses a model to score different subsets of features to finally select the best one. Forward selection is one of the Wrapper based methods, which starts from a null model with zero features and adds them greedily one at a time to maximize the model performance.

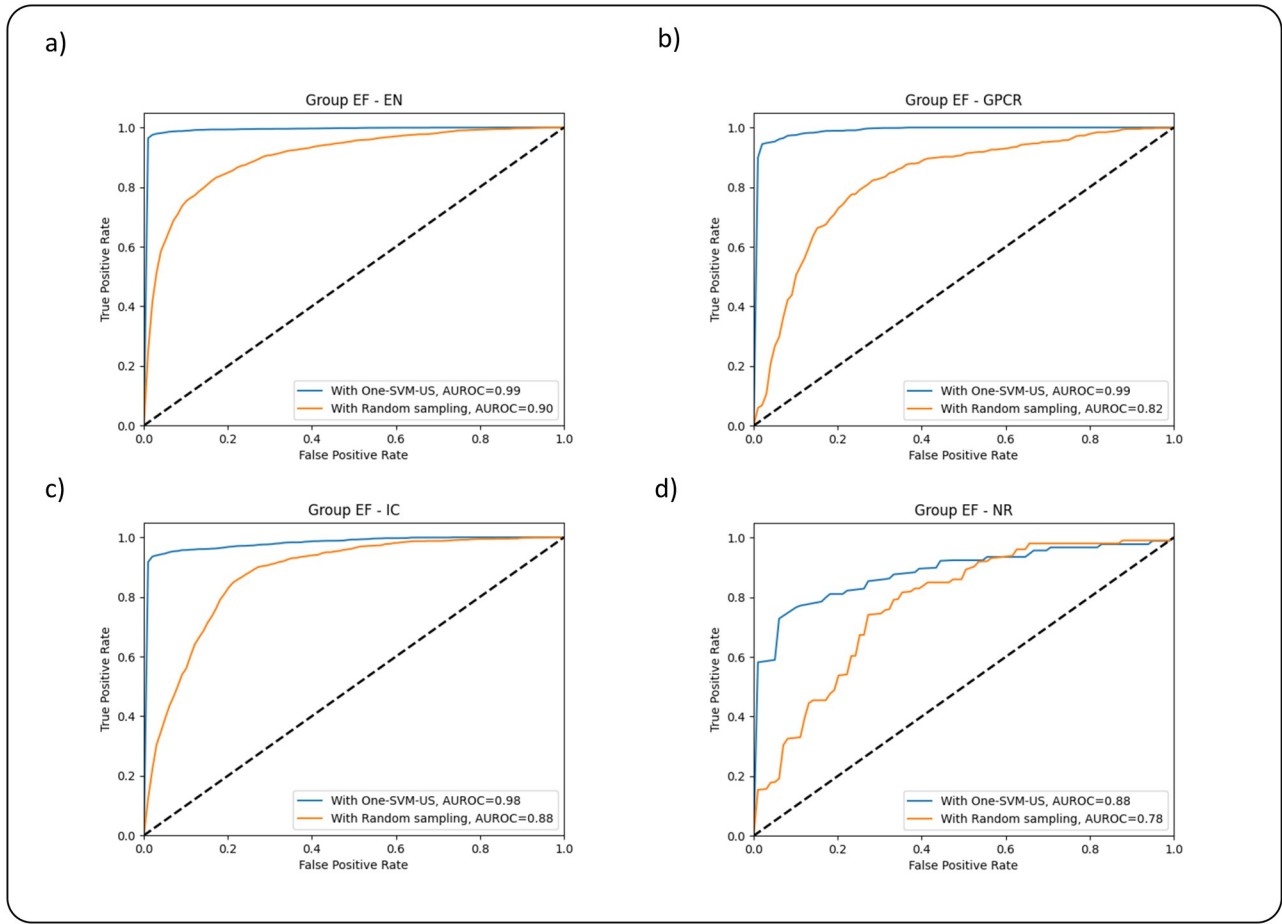

**Fig 3. ROC curves of the feature group EF using Random undersampling and One-SVM-US techniques on the datasets: (a) EN, (b) GPCR, (c) IC, and (d) NR.**

Here, we use the FFS-RF algorithm to find the optimal subset and maximize performance. Table 9 indicates the performance results of FFS-RF on the EN dataset in groups AB, CD, EF, and GHI. Table 9 shows ACC, AUROC, and AUPR metrics of the FFS-RF method which reduces the input features to the model. The worth of the FFS-RF is clearly observable; For the EN dataset, we just use 8 features instead of 676 features in group AB, 10 features instead of

**Table 9. The performance results of FFS-RF on the EN dataset.**

| Feature Combination | Number of Features | Number of Selected Features | ACC | AUROC | AUPR |
|---|---|---|---|---|---|
| AB (with drug features) | 676 | 8 | 1.0000 | 0.9910 | 0.9968 |
| CD (with drug features) | 661 | 10 | 0.9897 | 0.9903 | 0.9958 |
| EF (with drug features) | 504 | 7 | 0.9874 | 0.9903 | 0.9954 |
| GHI (with drug features) | 556 | 10 | 0.9965 | 0.9923 | 0.9976 |

**Table 10.  The performance results of FFS-RF on the datasets in group AB.**

| Feature Combination | Dataset | Number of Selected Features | ACC | AUROC | AUPR |
|---|---|---|---|---|---|
| **AB** (with drug features) | EN | 8 | 1.0000 | 0.9910 | 0.9968 |
| | GPCR | 4 | 0.9921 | 0.9854 | 0.9924 |
| | IC | 9 | 0.9695 | 0.9715 | 0.9769 |
| | NR | 4 | 0.9722 | 0.9217 | 0.9282 |

661 features in group CD, 7 features instead of 504 features in group EF, and 10 features instead of 556 features in group GHI. Moreover, the ACC of the FFS-RF method is 100%, 98%, 98%, and 99% in groups AB, CD, EF, and GHI, respectively. The AUROC and AUPR scores are approximately 0.99 in all four groups. In the case of the EN dataset, the feature groups AB and EF had the best and the worst model performance. So, we performed the FFS-RF method on the remaining datasets, i.e. GPCR, IC, and NR for these feature groups. The ACC, AUROC, and AUPR metrics are shown in Tables 10 and 11 for groups AB and EF, respectively. In group AB, the best feature dimensions selected by FFS-RF are 8, 10, 7, and 10, respectively, which ACC scores are 100%, 99%, 96%, and 97%. The AUROC values of group AB for FFS-RF are 0.9910, 0.9854, 0.9715, and 0.9217. In this group, 0.9968, 0.9924, 0.9769, and 0.9282 are obtained for the AUPR metric. We can see a similar pattern for group EF. Thus, FFS-RF is an effective method to avoid overfitting, improve prediction performance and reduce experimental cost.

## 6.5 Selection of predictor model

In this study, we focus on four classifiers: SVM, Random Forest (RF), MLP, and XGBoost. To evaluate these classifier models, we apply Cross Validation (CV) technique to select an appropriate predictor model for our problem. The results of the different predictive models are shown for the EN dataset in group AB in Table 12. To make an obvious comparison of prediction effects, the results are also demonstrated as a bar graph for the EN dataset in Fig 4. Comparison among the prediction results of the EN dataset from Table 12 reveals that the highest results of AUROC, AUPR, ACC, SEN, SPE, and F1 obtained by the XGBoost algorithm are 0.9920, 0.9975, 0.9901, 0.9947, 0.9967, and 0.9956, respectively. The overall prediction ACC of SVM, RF, MLP, and XGBoost is 0.8698, 0.9863, 0.8956, and 0.9901, respectively. The XGBoost ACC is 12%, 0.38%, and 9.45% higher than that obtained by SVM, RF, and MLP classifiers. The prediction performance of the XGBoost classifier is premier than the other three classifiers.

To make a better evaluation, we compare the DTI prediction performance of classifier models using the benchmark Yamanishi and Davis datasets. For each classifier, we use the balanced datasets with optimal features to predict DTIs. Table 13 provides a comparison of the XGBoost for SRX-DTI, as the best performing method, and RF, as the second-best performing

**Table 11.  The performance results of FFS-RF on the datasets in group EF.**

| Feature Combination | Dataset | Number of Selected Features | ACC | AUROC | AUPR |
|---|---|---|---|---|---|
| **EF** (with drug features) | EN | 7 | 0.9874 | 0.9903 | 0.9954 |
| | GPCR | 10 | 0.9737 | 0.9844 | 0.9893 |
| | IC | 9 | 0.9762 | 0.9647 | 0.9762 |
| | NR | 8 | 0.9444 | 0.8993 | 0.9296 |

**Table 12. The comparison of different ML algorithms on EN dataset in group AB.s.**

| Classifier | AUROC | AUPR | ACC | SEN | SPE | F1 |
|---|---|---|---|---|---|---|
| SVM | 0.9417 | 0.9544 | 0.8698 | 0.7898 | 0.9620 | 0.8629 |
| RF | 0.9910 | 0.9968 | 0.9863 | 0.9965 | 0.9967 | 0.9965 |
| MLP | 0.9586 | 0.9661 | 0.8956 | 0.8534 | 0.9488 | 0.8944 |
| XGBoost | 0.9920 | 0.9975 | 0.9901 | 0.9947 | 0.9967 | 0.9956 |

method under the 5-Fold CV on four datasets in groups AB and EF. Table 14 also reports the AUROC values under the 5-Fold CV on the Davis dataset. Average AUROC (Mean) values and standard deviation (Std) are also given in Tables 13 and 14 for each classifier model. These results indicate that the XGBoost outperforms other methods in different folds. Therefore, we select the XGBoost classifier as a classification algorithm to predict DTIs. Most of the classifiers pose low standard deviations which reveals our proposed model is a noise-resistant ML method and it does not depend on the classifier and dataset very much. Eventually, we can see the acceptable performance in most of the classifiers.

## 6.6 Comparison with other methods

During the last decade, different machine learning frameworks have been proposed to predict DTIs. Some of the proposed methods use feature selection techniques and some of those do not use feature selection. Most of the studies (as well as our approach) have used the dataset proposed by Yamanishi et al. [38] to assess the prediction ability of the proposed methods. To evaluate the effectiveness of our method, we consider six drug–target methods under the AUROC values for the same dataset under the 5-fold CV. In the following, we compare the AUROC of the SRX-DTI model with the other state-of-the-art methods proposed by Mousavian et al. [26], Li et al. [70], Meng et al. [71], Wang et al. [29], Mahmud et al. [64], Wang et al. [28], and Mahmud et al. [6]. The AUROCs generated by these models are listed in Table 15. As seen in the table, the AUROC of the proposed model is superior in comparison with the AUROC of other methods in all the datasets.

Average AUROC values of SRX-DTI on EN, GPCR, IC, and NR are 0.9920, 0.9880, 0.9788, and 0.9329, respectively. It should be considered that most of the existing models are without a

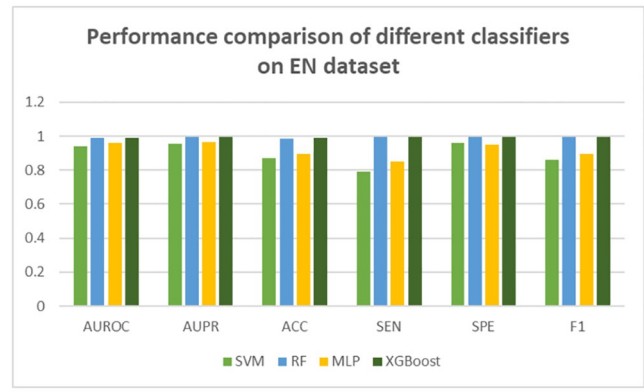

**Fig 4. Performance comparison of different feature selection techniques on EN dataset in group AB.**

**Table 13. Comparison of AUROC values under the 5-Fold cross-validation on four datasets in groups AB and EF.**

| Dataset | | Feature groups | | | | | | | |
|---|---|---|---|---|---|---|---|---|---|
| | | AB | | | | EF | | | |
| EN | Fold | SVM | RF | MLP | XGBOOST | SVM | RF | MLP | XGBOOST |
| | 1 | 0.9363 | 0.9921 | 0.9562 | 0.9949 | 0.9343 | 0.9925 | 0.9160 | 0.9952 |
| | 2 | 0.9563 | 0.9945 | 0.9655 | 0.9959 | 0.9177 | 0.9948 | 0.9180 | 0.9961 |
| | 3 | 0.9387 | 0.9978 | 0.9562 | 0.9978 | 0.9162 | 0.9975 | 0.9169 | 0.9972 |
| | 4 | 0.9510 | 0.9983 | 0.9633 | 0.9987 | 0.9284 | 0.9933 | 0.9241 | 0.9953 |
| | 5 | 0.9395 | 0.9950 | 0.9658 | 0.9958 | 0.9173 | 0.9938 | 0.9132 | 0.9936 |
| | Mean | 0.9417 | 0.9910 | 0.9586 | 0.9920 | 0.9211 | 0.9903 | 0.9158 | 0.9910 |
| | Std | 0.0078 | 0.0023 | 0.0043 | 0.0014 | 0.0073 | 0.0017 | 0.0036 | 0.0012 |
| GPCR | Fold | SVM | RF | MLP | XGBOOST | SVM | RF | MLP | XGBOOST |
| | 1 | 0.8122 | 0.9794 | 0.8284 | 0.9880 | 0.8700 | 0.9873 | 0.8145 | 0.9913 |
| | 2 | 0.7483 | 0.9977 | 0.7683 | 0.9990 | 0.8960 | 0.9894 | 0.8741 | 0.9923 |
| | 3 | 0.8342 | 0.9964 | 0.8301 | 0.9978 | 0.8688 | 0.9844 | 0.8254 | 0.9918 |
| | 4 | 0.7703 | 0.9855 | 0.7808 | 0.9896 | 0.8864 | 0.9924 | 0.8568 | 0.9944 |
| | 5 | 0.7825 | 0.9902 | 0.8000 | 0.9891 | 0.8850 | 0.9864 | 0.8452 | 0.9914 |
| | Mean | 0.7883 | 0.9854 | 0.8003 | 0.9880 | 0.8795 | 0.9844 | 0.8418 | 0.9881 |
| | Std | 0.0304 | 0.0068 | 0.0248 | 0.0047 | 0.0104 | 0.0027 | 0.0214 | 0.0011 |
| IC | Fold | SVM | RF | MLP | XGBOOST | SVM | RF | MLP | XGBOOST |
| | 1 | 0.8742 | 0.9645 | 0.8670 | 0.9822 | 0.8056 | 0.9736 | 0.8063 | 0.9862 |
| | 2 | 0.8842 | 0.9857 | 0.8537 | 0.9852 | 0.8012 | 0.9726 | 0.8197 | 0.9868 |
| | 3 | 0.8698 | 0.9626 | 0.8364 | 0.9730 | 0.7924 | 0.9674 | 0.8320 | 0.9859 |
| | 4 | 0.8818 | 0.9731 | 0.8559 | 0.9843 | 0.8033 | 0.9723 | 0.8170 | 0.9827 |
| | 5 | 0.8820 | 0.9826 | 0.8594 | 0.9883 | 0.8127 | 0.9579 | 0.7935 | 0.9781 |
| | Mean | 0.8762 | 0.9715 | 0.8528 | 0.9788 | 0.8008 | 0.9647 | 0.8118 | 0.9796 |
| | Std | 0.0055 | 0.0093 | 0.0101 | 0.0052 | 0.0066 | 0.0058 | 0.0130 | 0.0032 |
| NR | Fold | SVM | RF | MLP | XGBOOST | SVM | RF | MLP | XGBOOST |
| | 1 | 0.8210 | 0.9321 | 0.8210 | 0.9506 | 0.8981 | 0.9105 | 0.8858 | 0.8673 |
| | 2 | 0.8204 | 0.9195 | 0.7616 | 0.9288 | 0.8731 | 0.9009 | 0.7678 | 0.9319 |
| | 3 | 0.7608 | 0.9352 | 0.8164 | 0.9506 | 0.8812 | 0.9398 | 0.7670 | 0.9475 |
| | 4 | 0.8642 | 0.9583 | 0.8549 | 0.9738 | 0.8457 | 0.8765 | 0.8179 | 0.8364 |
| | 5 | 0.7817 | 0.8777 | 0.7539 | 0.8777 | 0.8638 | 0.8854 | 0.7678 | 0.8483 |
| | Mean | 0.8090 | 0.9217 | 0.8007 | 0.9329 | 0.8695 | 0.8993 | 0.8000 | 0.8837 |
| | Std | 0.0357 | 0.0266 | 0.0383 | 0.0326 | 0.0175 | 0.0220 | 0.0466 | 0.0450 |

feature selection phase [26,28,29,70,71]. Training the model with more features can lead to overfitting and reduce the power of generalization in the model. Whereas we can achieve the AUROC of 0.9920 in group AB by using just eight features instead of using all 676 features. This is significantly valuable in terms of computational cost. Moreover, our balancing method superlatively addresses the imbalance problem in the datasets, and feature selection techniques select an optimal subset of features for five datasets. Ultimately, the XGBoost classifier is so scalable that can perform better in comparison with other classifiers for identifying the new DTIs.

## 7. Conclusion

The identification of drug-target interactions through experimentation is a costly and time-consuming process. Therefore, the development of computational methods for identifying

**Table 14. Comparison of AUROC values under the 5-Fold cross-validation on Davis dataset.**

| Dataset | | Feature groups | | | | | | | |
|---|---|---|---|---|---|---|---|---|---|
| | | AB | | | | CD | | | |
| Davis | Fold | SVM | RF | MLP | XGBOOST | SVM | RF | MLP | XGBOOST |
| | 1 | 0.8926 | 0.9750 | 0.8715 | 0.9790 | 0.9327 | 0.9807 | 0.9019 | 0.9883 |
| | 2 | 0.9055 | 0.9752 | 0.8630 | 0.9798 | 0.9514 | 0.9875 | 0.9289 | 0.9877 |
| | 3 | 0.8893 | 0.9815 | 0.8599 | 0.9846 | 0.9185 | 0.9794 | 0.8853 | 0.9868 |
| | 4 | 0.9089 | 0.9832 | 0.8909 | 0.9852 | 0.9330 | 0.9861 | 0.9057 | 0.9891 |
| | 5 | 0.9005 | 0.9839 | 0.8941 | 0.9813 | 0.9295 | 0.9843 | 0.9206 | 0.9874 |
| | Mean | 0.8976 | 0.9769 | 0.8742 | 0.9786 | 0.9319 | 0.9797 | 0.9070 | 0.9839 |
| | Std | 0.0074 | 0.0039 | 0.0142 | 0.0025 | 0.0106 | 0.0031 | 0.0152 | 0.0008 |
| | | EF | | | | GHI | | | |
| | Fold | SVM | RF | MLP | XGBOOST | SVM | RF | MLP | XGBOOST |
| | 1 | 0.9095 | 0.9626 | 0.9054 | 0.9789 | 0.8818 | 0.9581 | 0.8671 | 0.9684 |
| | 2 | 0.9052 | 0.9698 | 0.9129 | 0.9776 | 0.8926 | 0.9683 | 0.8878 | 0.9741 |
| | 3 | 0.8988 | 0.9703 | 0.9030 | 0.9833 | 0.9008 | 0.9686 | 0.8809 | 0.9715 |
| | 4 | 0.9002 | 0.9694 | 0.8985 | 0.9814 | 0.8987 | 0.9677 | 0.8861 | 0.9752 |
| | 5 | 0.8919 | 0.9635 | 0.8902 | 0.9766 | 0.9096 | 0.9715 | 0.8983 | 0.9776 |
| | Mean | 0.8986 | 0.9643 | 0.8995 | 0.9756 | 0.8948 | 0.9636 | 0.8830 | 0.9696 |
| | Std | 0.0060 | 0.0034 | 0.0075 | 0.0025 | 0.0093 | 0.0046 | 0.0102 | 0.0032 |

**Table 15. Comparison of proposed model with existing methods on four datasets.**

| Dataset | Mousavian et al. [26] | Li et al. [70] | Meng et al. [71] | Wang et al. [29] | Mahmud et al. [64] | Wang et al. [28] | Mahmud et al. [6] | Proposed method |
|---|---|---|---|---|---|---|---|---|
| EN | 0.9480 | 0.9288 | 0.9773 | 0.9150 | 0.9808 | 0.9172 | 0.9656 | 0.9920 |
| IC | 0.8890 | 0.9171 | 0.9312 | 0.8900 | 0.9727 | 0.8827 | 0.9612 | 0.9880 |
| GPCR | 0.8720 | 0.8856 | 0.8677 | 0.8450 | 0.9390 | 0.8557 | 0.9249 | 0.9788 |
| NR | 0.8690 | 0.9300 | 0.8778 | 0.7230 | 0.9198 | 0.7531 | 0.8652 | 0.9329 |

interactions between drugs and target proteins has become a critical step in reducing the search space for laboratory experiments. In this work, we proposed a novel framework for predicting drug-target interactions. Our approach is unique in that we use a variety of descriptors for target proteins. We implement the One-SVM-US technique to address unbalanced data. The most important advantage of the proposed method is developing the FFS-RF algorithm to find an optimal subset of features to reduce computational cost and improve prediction performance. We also compare the performance of four classifiers on balanced datasets with optimal features, ultimately selecting the XGBoost classifier to predict DTIs in our model. We then employ the XGBoost classifier to predict DTIs on five benchmark datasets. Our SRX-DTI model achieved good prediction results, which showed that the proposed method outperforms other methods to predict DTIs.

The only limitation of this work can be the necessity of feature engineering in comparison with deep learning methods. However, the feature selection technique can also be considered a knowledge discovery tool that provides an understanding of the problem through the analysis of the most relevant features. On the other side, deep neural networks (DNNs) require large amounts of data to learn parameters, but our proposed model work on small data. This research showed that our robust framework is capable of capturing more potent and

informative features among massive features. Furthermore, the proposed framework poses resistance against noise and it is a data-independent machine learning method.

## Author Contributions

**Conceptualization:** Jamshid Pirgazi.

**Data curation:** Jamshid Pirgazi, Ali Ghanbari Sorkhi.

**Formal analysis:** Hakimeh Khojasteh, Jamshid Pirgazi, Ali Ghanbari Sorkhi.

**Investigation:** Hakimeh Khojasteh, Jamshid Pirgazi, Ali Ghanbari Sorkhi.

**Methodology:** Hakimeh Khojasteh.

**Project administration:** Jamshid Pirgazi.

**Software:** Hakimeh Khojasteh.

**Supervision:** Jamshid Pirgazi.

**Validation:** Hakimeh Khojasteh.

**Visualization:** Hakimeh Khojasteh, Ali Ghanbari Sorkhi.

**Writing – original draft:** Hakimeh Khojasteh.

**Writing – review & editing:** Hakimeh Khojasteh, Jamshid Pirgazi, Ali Ghanbari Sorkhi.

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
