## [Decision Letter · Decision Letter 0]

27 Mar 2023

PONE-D-23-03653Improving prediction of drug-target interactions based on fusing multiple features with data balancing and feature selection techniquesPLOS ONE

Dear Dr. Pirgazi,

Thank you for submitting your manuscript to PLOS ONE. After careful consideration, we feel that it has merit but does not fully meet PLOS ONE’s publication criteria as it currently stands. Therefore, we invite you to submit a revised version of the manuscript that addresses the points raised during the review process.

We look forward to receiving your revised manuscript.

Kind regards,

Prabina Kumar Meher, Ph.D.

Academic Editor

PLOS ONE

Journal Requirements:

2. We note that you have stated that you will provide repository information for your data at acceptance. Should your manuscript be accepted for publication, we will hold it until you provide the relevant accession numbers or DOIs necessary to access your data. If you wish to make changes to your Data Availability statement, please describe these changes in your cover letter and we will update your Data Availability statement to reflect the information you provide."

3. Please upload a new copy of Figure 1 as the detail is not clear. Please follow the link for more information: " ext-link-type="uri" xlink:type="simple">https://blogs.plos.org/plos/2019/06/looking-good-tips-for-creating-your-plos-figures-graphics/"
" ext-link-type="uri" xlink:type="simple">https://blogs.plos.org/plos/2019/06/looking-good-tips-for-creating-your-plos-figures-graphics/"

Reviewers' comments:

Reviewer's Responses to Questions

**Comments to the Author**

1. Is the manuscript technically sound, and do the data support the conclusions?

Reviewer #1: Yes

Reviewer #2: Partly

Reviewer #3: Partly

2. Has the statistical analysis been performed appropriately and rigorously? 

Reviewer #1: Yes

Reviewer #2: Yes

Reviewer #3: Yes

3. Have the authors made all data underlying the findings in their manuscript fully available?

Reviewer #1: Yes

Reviewer #2: Yes

Reviewer #3: Yes

4. Is the manuscript presented in an intelligible fashion and written in standard English?

Reviewer #1: Yes

Reviewer #2: Yes

Reviewer #3: No

5. Review Comments to the Author

Reviewer #1: This article proposes a new drug-target mutual prediction framework, which is called SRX-DTI. This method extracts the data of proteins and drugs, and proposes a new algorithm One-SVM-US to balance the data set, and then develops an algorithm called FFS-RF to predict the performance of the algorithm, and finally chooses to use XGBoost classification The device recognizes DTIs. The proposed method is compared with existing classical methods on standard datasets and defined subsets, and the results show that the proposed method has higher accuracy. However, before this manuscript is accepted, authors should address the following issues:

(1)In the manuscript, the author custom-divided the standard dataset into four subsets AB/CD/EF/GHI. It is recommended that the author provide the basis and method for dividing the data set in order to increase the credibility of the experiment in the article.

(2) The paper proposes a new algorithm One-SVM-US to balance the dataset, and the pseudocode in the article is worthy of praise. It is recommended to make a more detailed description in this section to increase the credibility of the experimental comparison of the article.

(3) It is recommended to describe in detail the method used by the FFR-RF algorithm in Section 6.4 when selecting feature dimensions.

(4) It is recommended to introduce new data sets for comparative experiments.

(5) The overall flowchart of the article is distorted after enlargement, and it is recommended that the author increase the clarity of the overall flowchart.

(6) The cited literature is insufficient, and appropriate addition of relevant literature will make the article more convincing, such as PMID 36088547, 34965582, 34456656, etc.

(7) The link https://github.com/Khojasteh-hb/SRX-DTI in the article is not available.Authors are advised to provide readers with all the data and software code on which the conclusion of the paper depends.

(8) In terms of writing technique, it is recommended to improve the condensation of sentences (especially condensed abstracts) and improve the continuity before and after certain chapters.

Reviewer #2: This paper proposes a new drug target interaction prediction framework called SRX-DTI. In this method, the protein sequence information and drug sequence information in the public dataset are selected to propose an algorithm called One-SVM-USD for the balanced dataset, and then the FFS-RF algorithm is developed to predict the algorithm performance, and XGBoost is selected for classification after experimental comparison. Finally, the overall performance is verified experimentally, and the results show that the framework performs well. However, before the manuscript is accepted, authors are advised to make improvements in the following areas:

(1) It is suggested that the authors provide the basis and method for dividing the four subsets of AB/CD/EF/GHI.

(2) This paper proposes a new algorithm One-SVM-US to balance the dataset, suggesting that the authors describe its innovation points in detail.

(3) The 08 data set is used in the article, and it is recommended that the author introduce a new data set.

(4) Authors are advised to improve the flowchart framework and improve its clarity.

(5) The link https://github.com/Khojasteh-hb/SRX-DTI in the article is not available.

It is recommended that authors provide more accurate and realistic code and data.

(6) In terms of writing techniques, it is recommended to improve the grammar and improve the logic of the chapters.

(7)It is recommended to increase the diversity of the table.

Reviewer #3: The author reported a computational method, called SRX-DTI. They proposed FFS-RF algorithm and present One-SVM-US technique. The results show superior performance in predicting DTIs. Besides, I still have some questions need to be solved.

1. In Abstract, the representation needs to be revised. I advised the author to read some related articles, to learn how the other authors represent. Generally, the logic of the article is not clear enough, need to be improved.

2. In Introduction, I know the topic of this paper is drug-target interaction. But the authors talk about Protein-protein interaction, It’s no problem. But I did not know the relationship between this paragraph. The logical needs to be sorted out.

3. In feature extraction section, the author proposed nine protein features, and construct for group, I want to know why these features only use four subsets? How about five groups? Or six?.. It needs to be clear expression or further talk about this reason.

4. I want to know One-SVM-US. What the full name is?

5. The picture in the text is too blurred. The dpi can be set more than 300.

6. if author have further extensive experiment.

7. so what’s the conclusion of this whole paper? And which the limitation of this work. Please show me.

In conclusion, I think this manuscript need to be large revised (mainly about the expression). And improve the English expression.

6. PLOS authors have the option to publish the peer review history of their article (what does this mean?). If published, this will include your full peer review and any attached files.

Reviewer #1: No

Reviewer #2: No

Reviewer #3: No

---

## [Author Response · Author response to Decision Letter 0]

10 May 2023

Dear Editor-in-Chief,

We wish to thank you, the associate editor, and the reviewers for the comments we received on the attached paper, and also thank you for giving us the opportunity to revise the manuscript again. We hereby submit a revised version of the paper. The detailed responses to reviewers’ comments are listed as follows, where reviewers’ comments are written in bold with our reply in a Calibri font. Furthermore, due to several new simulations and experimental results comparisons, Mr. Ali Ghanbari Sorkhi, has been added as a new co-author of this paper. We hope that the modified version is acceptable and we look forward to your kind recommendations.

Best regards,

The Authors

Reviewer Comments:

Response to Comments from Reviewer 1

This article proposes a new drug-target mutual prediction framework, which is called SRX-DTI. This method extracts the data of proteins and drugs, and proposes a new algorithm One-SVM-US to balance the data set, and then develops an algorithm called FFS-RF to predict the performance of the algorithm, and finally chooses to use XGBoost classification The device recognizes DTIs. The proposed method is compared with existing classical methods on standard datasets and defined subsets, and the results show that the proposed method has higher accuracy. However, before this manuscript is accepted, authors should address the following issues: 

We greatly appreciate the reviewer’s efforts and have made our best efforts to respond to all concerns raised by the review to the extent possible. We have made the following revisions accordingly.

(1) In the manuscript, the author custom-divided the standard dataset into four subsets AB/CD/EF/GHI. It is recommended that the author provide the basis and method for dividing the data set in order to increase the credibility of the experiment in the article.

Thanks for your ingenious comments and review. In the revised version, we have revised the section "Feature extraction methods" and explained the basis and method for dividing the data set. (Please see the section of Feature extraction methods technique, page 4, highlighted text.)

(2) The paper proposes a new algorithm One-SVM-US to balance the dataset, and the pseudocode in the article is worthy of praise. It is recommended to make a more detailed description in this section to increase the credibility of the experimental comparison of the article.

Thanks for the comments, in the revised version, we have explained the One-SVM-US algorithm more. We have also provided more details of its pseudocode. (Please see the section of Data balancing technique, pages 10 and 11, highlighted text.)

(3) It is recommended to describe in detail the method used by the FFR-RF algorithm in Section 6.4 when selecting feature dimensions.

Thanks for your valuable comment. In the revised version, we have added more detail for this algorithm. (Please see the section of Feature selection technique, page 11, highlighted text.)

(4) It is recommended to introduce new data sets for comparative experiments.

Thanks for the suggestion. We also evaluated our method on the Davis dataset and provided new comparative results. (Please see the subsection of Drug–target datasets, page 4, highlighted text, and the subsection of The influence of the data balancing techniques, pages 16 and 17, highlighted text and also, page 20, Table 14.)

(5) The overall flowchart of the article is distorted after enlargement, and it is recommended that the author increase the clarity of the overall flowchart.

Thanks for the comments. In revised version, we have improved the quality of flowchart.

(6) The cited literature is insufficient, and appropriate addition of relevant literature will make the article more convincing, such as PMID 36088547, 34965582, 34456656, etc.

Thanks for the comments. In the revised version, we have added these references. (Please see the section of Introduction, page 3, second highlighted paragraph.)

(7) The link https://github.com/Khojasteh-hb/SRX-DTI in the article is not available. Authors are advised to provide readers with all the data and software code on which the conclusion of the paper depends.

Thanks for the comments. We checked the link and it is okay. Also, the code source codes as well as the used datasets have been uploaded on GitHub. 

(8) In terms of writing technique, it is recommended to improve the condensation of sentences (especially condensed abstracts) and improve the continuity before and after certain chapters.

Thanks for the comments, we carefully studied the paper and we completely improved the condensation of sentences and the continuity before and after certain chapters.

 Response to Comments from Reviewer 2

This paper proposes a new drug target interaction prediction framework called SRX-DTI. In this method, the protein sequence information and drug sequence information in the public dataset are selected to propose an algorithm called One-SVM-USD for the balanced dataset, and then the FFS-RF algorithm is developed to predict the algorithm performance, and XGBoost is selected for classification after experimental comparison. Finally, the overall performance is verified experimentally, and the results show that the framework performs well. However, before the manuscript is accepted, authors are advised to make improvements in the following areas: 

We greatly appreciate the reviewer’s efforts to carefully review the paper and the valuable suggestions offered. We have made the following revisions accordingly.

 (1) It is suggested that the authors provide the basis and method for dividing the four subsets of AB/CD/EF/GHI.

Thanks for your ingenious comments and review. In the revised version, we have revised the section "Feature extraction methods" and explained the basis and method for dividing the data set. (Please see the section of Feature extraction methods technique, page 4, highlighted text.)

(2) This paper proposes a new algorithm One-SVM-US to balance the dataset, suggesting that the authors describe its innovation points in detail.

Thanks for the comments, in the revised version, we have explained the One-SVM-US algorithm more. We have also provided more details of its pseudocode. (Please see the section of Data balancing technique, page 10 and 11, highlighted text.)

(3) The 08 data set is used in the article, and it is recommended that the author introduce a new data set.

Thanks for the suggestion. We also evaluated our method on the Davis dataset and provided new comparative results. (Please see the subsection of Drug–target datasets, page 4, highlighted text, and the subsection of The influence of the data balancing techniques, pages 16 and 17, highlighted text and also, page 20, Table 14.)

(4) Authors are advised to improve the flowchart framework and improve its clarity.

Thanks for the comments. In revised version, we have improved the quality of flowchart.

(5) The link https://github.com/Khojasteh-hb/SRX-DTI in the article is not available.

It is recommended that authors provide more accurate and realistic code and data.

Thanks for the comments. We checked the link and it is okay. Also, the code source codes as well as the used datasets have been uploaded on GitHub. 

(6) In terms of writing techniques, it is recommended to improve the grammar and improve the logic of the chapters.

Thanks for the comments, we carefully studied the paper and we completely improved the grammar and the continuity before and after certain chapters.

(7) It is recommended to increase the diversity of the table.

Thanks for the suggestion. We increased the diversity of the table and added tables 13 and 14, which report the AUROC values under the 5-Fold cross-validation.

Response to Comments from Reviewer 3

The author reported a computational method, called SRX-DTI. They proposed FFS-RF algorithm and present One-SVM-US technique. The results show superior performance in predicting DTIs. Besides, I still have some questions need to be solved.

We greatly appreciate the reviewer’s efforts to carefully review the paper and the valuable suggestions offered. We have made the following revisions accordingly.

1. In Abstract, the representation needs to be revised. I advised the author to read some related articles, to learn how the other authors represent. Generally, the logic of the article is not clear enough, need to be improved.

Thanks for the comments. We studied related articles and we have improved the abstract.

2. In Introduction, I know the topic of this paper is drug-target interaction. But the authors talk about Protein-protein interaction, It’s no problem. But I did not know the relationship between this paragraph. The logical needs to be sorted out.

Thanks for the comments, we carefully reviewed the paper and we completely improved the condensation of sentences and the relationship between those paragraphs.

3. In feature extraction section, the author proposed nine protein features, and construct for group, I want to know why these features only use four subsets? How about five groups? Or six? It needs to be clear expression or further talk about this reason.

Thanks for your ingenious comments and review. In the revised version, we have revised the section "Feature extraction methods" and explained the basis and method for dividing the data set. (Please see the section of Feature extraction methods technique, page 4, highlighted text.)

4. I want to know One-SVM-US. What the full name is?

Thanks for your comment. The full name of One-SVM-US is UnderSampling by One-class Support Vector Machine. In the revised version, I have added the full name in the first mention of this name. (Please see the section of Introduction, page 3, third highlighted paragraph.)

5. The picture in the text is too blurred. The dpi can be set more than 300.

Thanks for the comments. In revised version, we have improved the quality of figures.

6. if author have further extensive experiment.

Thanks for the suggestion. We also evaluated our method on the Davis dataset and provided new comparative results. (Please see the subsection of Drug–target datasets, page 4, highlighted text, and the subsection of The influence of the data balancing techniques, pages 16 and 17, highlighted text and also, page 20, Table 14.)

7. so what’s the conclusion of this whole paper? And which the limitation of this work. Please show me.

Thanks for your ingenious comments and review. In the revised version, we have revised the conclusion section and mentioned the limitation of this work.

In conclusion, I think this manuscript need to be large revised (mainly about the expression). And improve the English expression.

Thanks for your comments. In the revised version, we have completely improved the English expression and grammar.

---

## [Decision Letter · Decision Letter 1]

22 Jun 2023

Improving prediction of drug-target interactions based on fusing multiple features with data balancing and feature selection techniques

PONE-D-23-03653R1

Dear Dr. Pirgazi,

We’re pleased to inform you that your manuscript has been judged scientifically suitable for publication and will be formally accepted for publication once it meets all outstanding technical requirements.

Kind regards,

Prabina Kumar Meher, Ph.D.

Academic Editor

PLOS ONE

Reviewers' comments:

Reviewer's Responses to Questions

**Comments to the Author**

1. If the authors have adequately addressed your comments raised in a previous round of review and you feel that this manuscript is now acceptable for publication, you may indicate that here to bypass the “Comments to the Author” section, enter your conflict of interest statement in the “Confidential to Editor” section, and submit your "Accept" recommendation.

Reviewer #1: All comments have been addressed

Reviewer #2: All comments have been addressed

Reviewer #3: All comments have been addressed

2. Is the manuscript technically sound, and do the data support the conclusions?

Reviewer #1: Yes

Reviewer #2: Yes

Reviewer #3: Yes

3. Has the statistical analysis been performed appropriately and rigorously? 

Reviewer #1: Yes

Reviewer #2: Yes

Reviewer #3: Yes

4. Have the authors made all data underlying the findings in their manuscript fully available?

Reviewer #1: Yes

Reviewer #2: Yes

Reviewer #3: Yes

5. Is the manuscript presented in an intelligible fashion and written in standard English?

Reviewer #1: Yes

Reviewer #2: Yes

Reviewer #3: Yes

6. Review Comments to the Author

Reviewer #1: The author of this manuscript has fully answered my question and I have no further questions. Recommend journals to accept this version of the manuscript.

Reviewer #2: (No Response)

Reviewer #3: The author reported a computational method, called SRX-DTI. They proposed FFS-RF algorithm and present One-SVM-US technique. The results show superior performance in predicting DTIs.

After review, All the problem have been addressed.

7. PLOS authors have the option to publish the peer review history of their article (what does this mean?). If published, this will include your full peer review and any attached files.

Reviewer #1: No

Reviewer #2: **Yes: **Yu changqing

Reviewer #3: No

---

## [Editor Report · Acceptance letter]

26 Jul 2023

PONE-D-23-03653R1 

Improving prediction of drug-target interactions based on fusing multiple features with data balancing and feature selection techniques 

Dear Dr. Pirgazi:

I'm pleased to inform you that your manuscript has been deemed suitable for publication in PLOS ONE. Congratulations! Your manuscript is now with our production department. 

Kind regards, 

on behalf of

Dr. Prabina Kumar Meher 

Academic Editor

PLOS ONE